# Reproducibility of CT-Based Hepatocellular Carcinoma Radiomic Features across Different Contrast Imaging Phases: A Proof of Concept on SORAMIC Trial Data

**DOI:** 10.3390/cancers13184638

**Published:** 2021-09-16

**Authors:** Abdalla Ibrahim, Yousif Widaatalla, Turkey Refaee, Sergey Primakov, Razvan L. Miclea, Osman Öcal, Matthias P. Fabritius, Michael Ingrisch, Jens Ricke, Roland Hustinx, Felix M. Mottaghy, Henry C. Woodruff, Max Seidensticker, Philippe Lambin

**Affiliations:** 1The D-Lab, Department of Precision Medicine, GROW—School for Oncology, Maastricht University, 6200 MD Maastricht, The Netherlands; y.widaatalla@maastrichtuniversity.nl (Y.W.); S.primakov@maastrichtuniversity.nl (S.P.); h.woodruff@maastrichtuniversity.nl (H.C.W.); philippe.lambin@maastrichtuniversity.nl (P.L.); 2Department of Radiology and Nuclear Medicine, Maastricht University Medical Centre+, 6200 MD Maastricht, The Netherlands; razvan.miclea@mumc.nl (R.L.M.); felix.mottaghy@mumc.nl (F.M.M.); 3Division of Nuclear Medicine and Oncological Imaging, Department of Medical Physics, University Hospital of Liege and GIGA CRC-In Vivo Imaging, University of Liege, 4000 Liege, Belgium; rhustinx@ulg.ac.be; 4Department of Nuclear Medicine and Comprehensive Diagnostic Center Aachen (CDCA), University Hospital RWTH Aachen University, 52074 Aachen, Germany; 5Department of Diagnostic Radiology, Faculty of Applied Medical Sciences, Jazan University, Jazan 45142, Saudi Arabia; 6Department of Radiology, University Hospital, LMU Munich, 80336 Munich, Germany; osman.oecal@med.uni-muenchen.de (O.Ö.); Matthias.Fabritius@med.uni-muenchen.de (M.P.F.); Michael.Ingrisch@med.uni-muenchen.de (M.I.); jens.ricke@med.uni-muenchen.de (J.R.); Max.Seidensticker@med.uni-muenchen.de (M.S.)

**Keywords:** hepatocellular carcinoma, CT radiomics, domain translation, reproducibility

## Abstract

**Simple Summary:**

Radiomics has been reported to have potential for correlating with clinical outcomes. However, handcrafted radiomic features (HRFs)—the quantitative features extracted from medical images—are limited by their sensitivity to variations in scanning parameters. Furthermore, radiomics analyses require big data with good quality to achieve desirable performances. In this study, we investigated the reproducibility of HRFs between scans acquired with the same scanning parameters except for the imaging phase (arterial and portal venous phases) to assess the possibilities of merging scans from different phases or replacing missing scans from a phase with other phases to increase data entries. Additionally, we assessed the potential of ComBat harmonization to remove batch effects attributed to this variation. Our results show that the majority of HRFs were not reproducible between the arterial and portal venous phases before or after ComBat harmonization. We provide a guide for analyzing scans of different imaging phases.

**Abstract:**

Handcrafted radiomic features (HRFs) are quantitative imaging features extracted from regions of interest on medical images which can be correlated with clinical outcomes and biologic characteristics. While HRFs have been used to train predictive and prognostic models, their reproducibility has been reported to be affected by variations in scan acquisition and reconstruction parameters, even within the same imaging vendor. In this work, we evaluated the reproducibility of HRFs across the arterial and portal venous phases of contrast-enhanced computed tomography images depicting hepatocellular carcinomas, as well as the potential of ComBat harmonization to correct for this difference. ComBat harmonization is a method based on Bayesian estimates that was developed for gene expression arrays, and has been investigated as a potential method for harmonizing HRFs. Our results show that the majority of HRFs are not reproducible between the arterial and portal venous imaging phases, yet a number of HRFs could be used interchangeably between those phases. Furthermore, ComBat harmonization increased the number of reproducible HRFs across both phases by 1%. Our results guide the pooling of arterial and venous phases from different patients in an effort to increase cohort size, as well as joint analysis of the phases.

## 1. Introduction

Recent decades have witnessed vast advances in computational power, artificial intelligence, and medical imaging techniques [1], which have provided a unique opportunity for transforming the abundant amounts of medical imaging into mineable quantitative data. This concept acquired much scientific attention recently, and a branch of medical imaging analysis—known as handcrafted radiomics—emerged as a result [2]. Handcrafted radiomic features (HRFs) are quantitative features extracted with high throughput from medical imaging, with its varying modalities. The hypothesis is that medical images carry more data than can be seen by trained human eyes, and that these data can be decoded using HRFs; i.e., correlations between HRFs and underlying biology could potentially exist [3]. Since the introduction of the field, many studies have reported on the potential of radiomic signatures to predict clinical endpoints, the majority of which were performed on computed tomography (CT) [4,5,6,7], magnetic resonance (MR) [8,9,10], and positron emission tomography (PET) scans [11,12].

Hepatocellular carcinoma (HCC) is the most common primary liver cancer, the fifth most common malignancy worldwide, and a leading cause of cancer-related mortality [13]. Different diagnostic approaches and treatment modalities are used clinically depending on the characteristics of the patient and the progression of the disease [14,15]. Contrast-enhanced computed tomography (CE-CT) scans are considered one of the main diagnostic tools for HCC. CE-CT can be acquired at different times following the injection of the contrast agent to acquire arterial, venous or late phase scans. Each phase shows specific characteristics for HCC lesions. However, there is still a clinical need for reliable non-invasive tools that could aid in diagnosing and devising individualized treatment plans for HCC patients. Several studies have investigated and reported on the potential of HRFs to aid clinical decision making in HCC patients [16,17,18,19]. 

While numerous studies have reported on the potential of HRFs in aiding clinical decision making on HCC and other diseases, several hurdles hindering the clinical translation of radiomic signatures to clinical decision support systems have been identified. These hurdles include the reproducibility of HRFs in test-retest studies, their sensitivity to variations in acquisition and reconstruction parameters of the scans, inter-observer variability, and the need for big data [20,21,22,23,24,25,26]. However, the need for big data in radiomics analysis necessitates the exploration of methods for combining and comparing retrospective medical imaging databases.

A number of studies have tried to address the issue of reproducibility of HRFs using ComBat harmonization [27,28,29,30]. ComBat harmonization is a method that was developed to remove the batch effects in gene expression arrays [31]. The studies that investigated the application of ComBat in radiomics analyses reported on the improvement in performance metrics of developed radiomic signatures after the application of ComBat compared to before, and recommended the use of the method. Other studies investigated the reproducibility of HRFs on phantom datasets acquired with different settings [32], or with a single parameter difference [33], and reported that the performance of ComBat is dependent on the data under study. These studies recommended a framework to assess the reproducibility of HRFs. However, to date, no study has reported on the agreement in HRFs across different phases, the potential of ComBat to remove the effects of different imaging phases from HRFs, which could allow the proper combination of phases in a single analysis, or the interchangeability of HRFs across phases to allow the use of different imaging scans per patient. Furthermore, no study has performed a reproducibility analysis for HRFs following ComBat harmonization on patients’ scans acquired with a single parameter difference.

We hypothesize that the time of acquisition after the injection of the contrast agents adds another level of complexity to be accounted for in the radiomics analysis, as HRFs might be affected by the appearance of contrast due to the variations in the distribution of the contrast within the lesions. As a proof of concept, we investigate the sensitivity of HRFs extracted from CE-CT scans depicting HCC acquired during the arterial and portal venous phases when all other acquisition and reconstruction parameters were fixed. Furthermore, we investigate the potential of the ComBat harmonization for domain translation of the HRFs extracted from these scans. Ultimately, we aim to (i) guide the identification of HRFs that can be used interchangeably between arterial and venous phase scans, which could increase the number of scans that can be included in a CE-CT based radiomics study; and (ii) identify the features that can be used in studies analyzing both phases simultaneously to maximize the information extracted from ROIs.

## 2. Materials and Methods

### 2.1. Patients and Imaging Data

The imaging data were originally collected for the European multicenter clinical trial (SORAMIC) [34]. Imaging data for 424 patients diagnosed with HCC (using cyto-histological criteria, radiologic criteria, or a combination of both) were obtained for the SORAMIC trial, of which 338 scans were available for analysis in this study. Scans that contained artifacts were considered of poor quality (n = 48). From the available 338 patients with both arterial and portal venous scans available, patients with scans that had any difference in the acquisition or reconstruction parameters, or lacked segmentations reviewed by an expert, were excluded. A total of 61 patients with 104 distinct lesions were finally included in this study (Figure 1). Scans included were acquired from different hospitals using different vendors and protocols. In total, 9 scanner models from 4 different imaging vendors, and a range of scanning parameters, were included, as shown in Table 1. The imaging analysis was approved by the University of Magdeburg institutional review board (IRB00006099, EudraCT no 2009-012576-27), and informed consent was obtained from all included patients. All methods were carried out in accordance with the relevant guidelines and regulations [35].

### 2.2. Segmentation and HRFs Extraction

The scans of a single patient were co-registered. The region of interest (ROI) was segmented on each scan while viewing both phases simultaneously and saved to both scans (Figure 2). The segmentations were performed using MIM software (MIM Software Inc., Cleveland, OH) by a medical doctor (Y.W.) with 2 years of experience in image segmentation, and revised by a radiologist (R.M.) with 15 years of experience in medical radiology.

HRFs were extracted from these ROIs using the software RadiomiX Discovery Toolbox (version, October 2019; https://www.radiomics.bio, accessed on 12 January 2021), which calculates HRFs compliant with the Imaging Biomarkers Standardization Initiative (IBSI) [36], in addition to others. Image intensities were binned with a binwidth of 25 Hounsfield Units (HUs) in order to reduce noise levels and to reduce texture matrix sizes, and therewith computation power, with no resampling or further preprocessing of the images. The description of the extracted HRFs was published previously [24].

### 2.3. ComBat Harmonization

The ComBat method employs empirical Bayes to estimate the effects of assigned batches on the data being harmonized. For HRFs, ComBat assumes that a feature value can be approximated by the equation:(1)$$Y_{ij}=\alpha +\beta X_{ij}+\gamma_{i}+\delta_{i}\varepsilon_{ij}$$
where α is the average value for HRF *Y**_ij_* for ROI j on scanner *i*; *X* is a design matrix of the biologic covariates that are known to affect the value of HRFs; *β* is the vector of regression coefficients corresponding to each biologic covariate; *γ**_i_* is the additive effect of scanner *i* on HRFs, *δ**_i_* is the multiplicative scanner effect, and *ε_ij_* is an error term, presupposed to be normally distributed with zero mean. Based on the values estimated, ComBat performs feature transformation as given by the formula:(2)$$Y_{ij}^{ComBat}=\frac{\left(Y_{ij}-\hat{\alpha }-\hat{\beta }X_{ij}-\gamma_{i}^{*}\right)}{\delta_{i}^{*}}+\hat{a}+\hat{\beta }X_{ij}$$
where $\hat{\alpha }$ and $\hat{\beta }$ are estimators of the parameters *α* and *β*, respectively; and $\gamma_{i}^{*}$ and $\delta_{i}^{*}$ are the empirical Bayes estimates for the parameters *γ**_i_* and *δ_i_*, respectively.

### 2.4. Statistical Analysis

All statistical analyses were performed using R language [37] on RStudio (V 3.6.3) [38]. To determine the reproducibility of HRFs, the concordance correlation coefficient (CCC) between the HRF values across the two phases was calculated [39] using the epiR package [40]. The CCC measures how concordant the values of a given HRF are, as well as the rank of each data point relative to the rest in each batch. HRFs with CCC > 0.9 were considered reproducible and could be interchangeably used between the arterial and venous phase CT scans. 

To assess the performance of ComBat, shape features and HRFs with (near) zero variance (HRFs that have the same value in 95% or more of the observations) were removed. The phase of the scan was assigned as the batch for ComBat harmonization. The CCC was calculated after ComBat application and the cutoff of CCC > 0.9 was applied to select the concordant HRFs. The correlation of concordant features with volume was assessed using Pearson correlation. Features that had a correlation coefficient > 0.85 were considered highly correlated. The analysis code used in this study can be found on: (https://github.com/AbdallaIbrahim/The-reproducibility-and-ComBatability-of-Radiomic-features, accessed on 6 September 2021).

## 3. Results

### 3.1. Patient Characteristics

The patients included (n = 61) had a median age of 66 years, were mainly male (n = 50, 81.9%) with cirrhotic livers (n = 56, 91.8%), and a minority (n = 11, 18.1%) had portal vein invasion. For more patient characteristics, see Table 2.

### 3.2. Extracted HRFs

A total of 167 original HRFs were extracted from each of the available 104 ROIs. These HRFs are divided into 11 feature families: Fractal (n = 3), Gray Level Co-occurence Matrix (GLCM; n = 26), Gray Level Distance Zone Matrix (GLDZM; n = 16), Gray Level Run Length Matrix (GLRLM; n = 15), Gray Level Size Zone Matrix (GLSZM, n = 16), Intensity Histogram (IH; n = 25), Local Intensity (LocInt, n = 2), Neighbouring Gray Level Dependence Matrix (NGLDM; n = 17), Neighbouring Gray Tone Difference Matrix (NGTDM, n = 5), Shape (n = 23), and Statistics (Stats, n = 19).

### 3.3. The Effects of Differences in Imaging Phase on the Reproducibility of HRFs

Out of the 167 extracted HRFs, 42 (25%) were reproducible (had a CCC > 0.9) across both phases (Figure 3a, shape features were not included to ease the comparison between figures). These HRFs were divided into Shape (n = 22), NGTDM (n = 1), NGLDM (n = 4), IH (n = 2), GLSZM (n = 4), GLRLM (n = 2) and GLDZM (n = 7). The remaining HRFs had a CCC ranging from −0.07 and 0.85, with a median of 0.39.

Of the concordant 22 shape features, 8 features were highly correlated with volume (R > 0.85), in addition to 1 feature from the NGLDM group (NGLDM_DN) and 2 features from the GLRLM group (GLRLM_RLN and GLRLM_GLN). The remaining features (31, 73.8%) had a correlation coefficient < 0.85.

### 3.4. The Effects of ComBat on the Reproducibility of HRFs

The application of ComBat harmonization to remove the batch effects attributed to the difference in time between contrast injection and scan acquisition resulted in a total of 44 (26.1%) reproducible HRFs; i.e., 2 extra HRFs became concordant following the application of ComBat: Stats_energy and GLDZM_HILDE (Figure 3b). The remaining 20 HRFs had a CCC > 0.9 before and after ComBat harmonization, in addition to the shape features (n = 22). The CCC of stats_energy increased from 0.8 to 0.95 following ComBat harmonization, and that of GLDZM_HILDE increased from 0.34 to 0.93.

The impact of ComBat on the CCC values had a wide range: 6 HRFs had an increment in CCC between 0.5 and 0.6; 42 HRFs had an increment in CCC between 0.1 and 0.49; 87 HRFs had an increment between 0 and 0.09; and 33 HRFs had a decrement in CCC between −0.001 and −0.06. Following ComBat harmonization, the number of highly correlated features with volume increased by one feature (Stats_energy). The concordant features before domain translation maintained their correlation with volume.

## 4. Discussion

In this study, we investigated the reproducibility of HCC CT-based HRFs across the arterial and portal venous imaging phases when all other scanning parameters were fixed, and whether ComBat harmonization improves the reproducibility of HRFs in such a scenario. Uniquely, this is the first manuscript to investigate the potential of ComBat to remove batch effects attributed to the differences in imaging phase from patient data with a single parameter difference between the compared/harmonized scans. Our results show that the majority of HRFs were significantly affected by the difference in imaging phases, and only a quarter of the total extracted number of HRFs were reproducible across both phases. Moreover, ComBat harmonization did not successfully harmonize the majority of HRFs, even though the differences between the batches compared were limited to the variations in imaging phase.

HRFs are calculated using mathematical formulas applied on the array of values representing the medical image [41]. Changes in the value of units in this array are expected to have an impact on the value calculated by the same formula. Therefore, changes in the scanning parameters are expected to affect the reproducibility of different HRFs variably. Aside from HRFs that are not reproducible in test-retest studies, the sensitivity of the remaining HRFs to the imaging phase can be justified by the increased radio-opaqueness and the resulting perfusion patterns of contrast within the ROI, and thus, changes in the image array values based on which the HRFs are calculated. As expected, statistics and intensity histogram features, which are simple HRFs based on a single voxel value (e.g., minimum or maximum intensity value) or the description of their distribution (e.g., mean or median intensity value), were found to be the most significantly affected families. On the other hand, also according to expectations, HRFs that do not depend on the intensity values, but the shape of the segmentation (shape features), were found to be reproducible across both phases, with the exception of the shape feature centroid distance, which is based on the distribution of intensity values around the geometric center of the ROI. The copying of segmentations and the inclusion of scans that were acquired identically in both phases allowed isolation of the effects of differences in imaging phases on HRFs. However, in scenarios where acquisition and/or reconstruction parameters, or the segmentation of the ROI changes, the reproducibility of HRFs is expected to be further impacted. This is also in line with what has been reported in a study that investigated the reproducibility of liver parenchyma and tumor HRFs extracted from two contrast-enhanced scans (one phase) taken within a 14 day interval [42]. Therefore, the reproducibility analysis based on the data under study should be an integral part of each radiomics study.

Our study sheds the light on the methodology of combining HRFs from different modalities, either for the purpose of combining different phases/modalities per patient or for combining different phases for different patients. For merging different modalities per patient, we show that a number of HRFs are reproducible across phases. Therefore, models that try to combine different imaging phases per patient are recommended to define which reproducible (test-retest) HRFs vary across the available phases and preselect those for further analysis. Another implication of our findings is allowing the combination of different imaging phases per patient (e.g, due to the lack of data) when only the reproducible HRFs across phases are extracted and compared between the different patients, regardless of the available imaging phase for each patient. This approach can significantly increase the number of data points in retrospective radiomics studies.

The correlation of radiomic features with the volume of the ROI has been considered one of the major points to be assessed in radiomics analysis, since some of the features were reported previously to be surrogates of volume [43]. In our analysis, we observed that the majority of the features identified as concordant (or domain-translatable with ComBat) between the arterial and venous CT scans was considerable, most of which were shape features. However, the majority of features were not found to be highly correlated with volume, which means that these features can decode additional information about the ROIs being investigated.

The number of features that had a CCC value higher than 0.9 was slightly higher after the application of ComBat on the HRFs extracted from the arterial and portal venous phases. ComBat successfully harmonized two additional HRFs compared to the number of concordant HRFs before domain translation. The majority of HRFs were not concordant across the phases even after the application of ComBat harmonization. The differences in ComBat performance per HRF (and feature families) are also expected, as in contrast to gene expression arrays, HRFs have different levels of complexity and are not expected to be uniformly affected by the batch defined for domain translation. The variant performance of ComBat on HRFs could be explained by the differences in the complexity of HRFs compared to gene expression arrays [21]. The findings are in line with the reproducibility studies that assessed the performance of ComBat on phantom scans, which reported that ComBat harmonization does not successfully harmonize all HRFs, and that its performance is dependent on variations between batches [32,33]. As a consequence, we recommend that the application of ComBat harmonization on HRFs follows a reproducibility analysis with reference values to assess its performance, as it is expected to vary with variations in the dataset batches being harmonized [21]. Other deep learning-based harmonization methods that have been recently investigated [44,45,46,47] might be more suitable for domain translation of images acquired in different phases. However, this is yet to be investigated.

While this study provides a proof of concept for the combination/replacement of different imaging phases, we speculate that the set of reproducible HRFs identified in this study is limited to HCC lesions extracted from scans acquired similarly to our dataset. Furthermore, the changes in reconstruction parameters (and sometimes acquisition parameters) between the two imaging phases in clinical routine significantly lowered the number of available scans to perform this analysis. Lastly, the reproducibility of the identified HRFs has to be investigated across different acquisition and reconstruction parameters. However, due to the lack of data, this was not performed. Nevertheless, this study serves as a guide for selecting and/or harmonizing reproducible HRFs in future radiomic studies that utilize contrast-enhanced imaging.

## 5. Conclusions

The majority of HRFs are significantly affected by changes in the imaging phase of the scan. Studies that investigate the potential of combining HRFs from different imaging phases or modalities must investigate the reproducibility and interoperability of the HRFs across the investigated phases for the lesions of interest. Furthermore, a number of HRFs can be interchangeably used between the arterial and portal venous phases, and these can be used to increase data points in retrospective imaging studies. ComBat harmonization increased the number of comparable CT-based HRFs across the arterial and portal venous imaging phases for HCC lesions by 1% in our dataset.

## Figures and Tables

**Figure 1 cancers-13-04638-f001:**
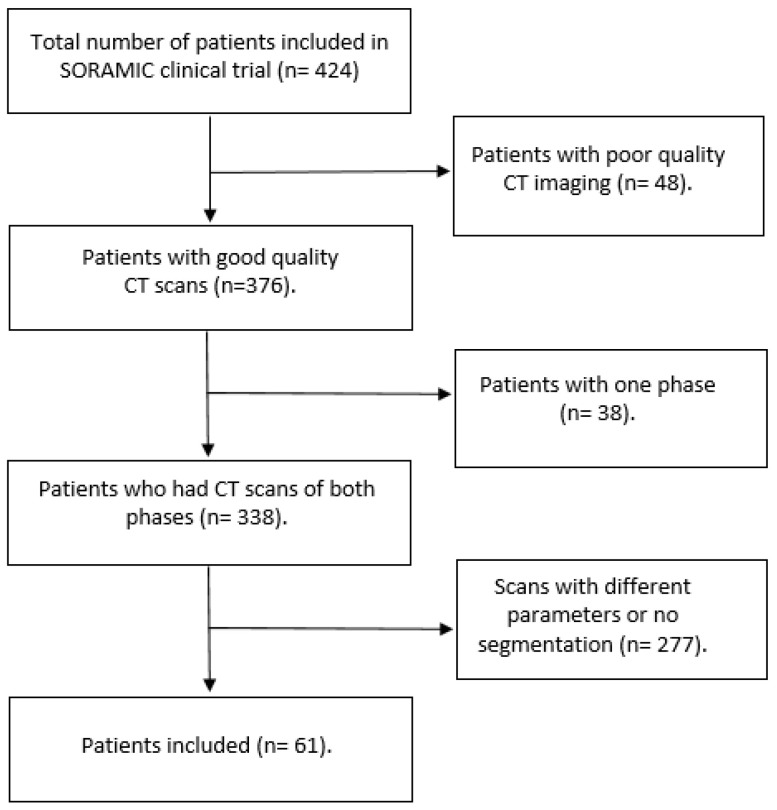
A flowchart showing the patient selection process.

**Figure 2 cancers-13-04638-f002:**
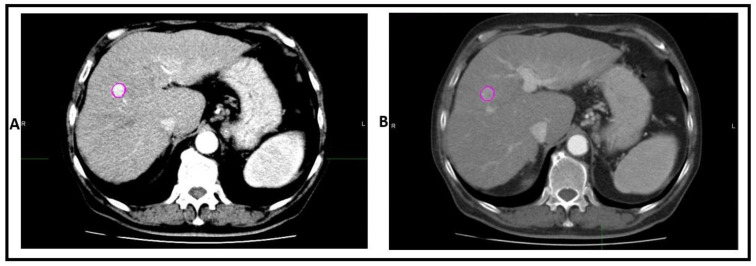
An example of ROI segmented in (**A**) the arterial phase and (**B**) the portal venous phase.

**Figure 3 cancers-13-04638-f003:**
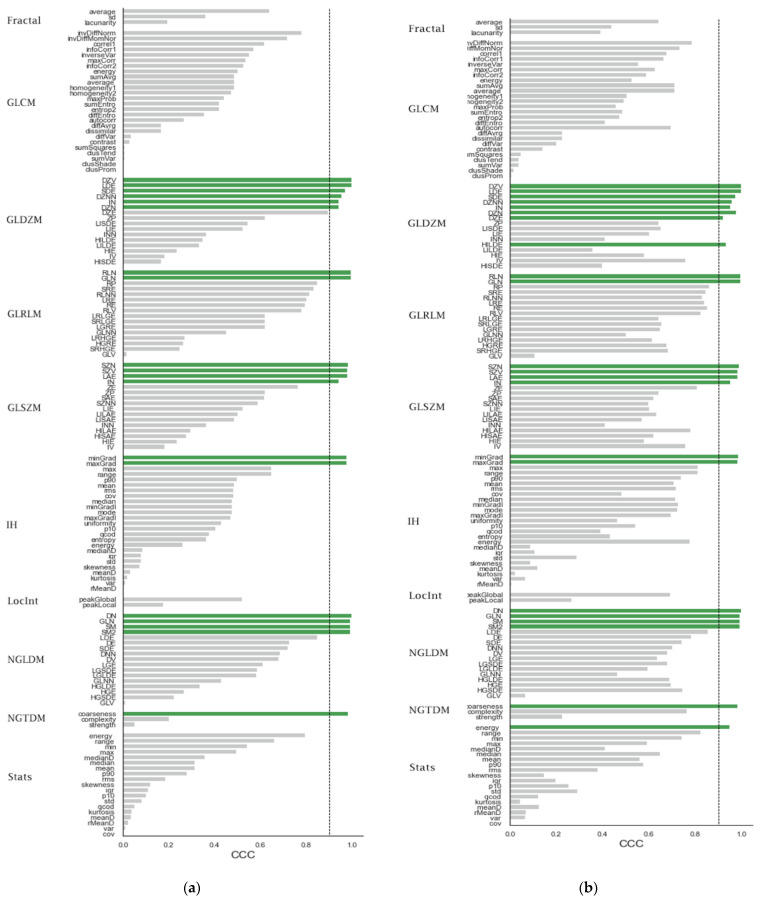
(**a**) The CCC values for the different HRFs before ComBat harmonization. (**b**) The CCC values for the different HRFs after ComBat harmonization.

**Table 1 cancers-13-04638-t001:** Acquisition and reconstruction parameters for the imaging dataset.

Manufacturer	Scanner Model	X-ray Tube Current (kV)	Exposure (mAs)	Convolution Kernels	Slice Thickness (mm)	Pixel Spacing (mm^2^)
TOSHIBA	Aquilion	50–360	2–300	FC13	1–5	0.39 × 0.39 − 0.98 × 0.98
	Aquilion PRIME					
Philips	Brilliance 64			B		
GE	Discovery CT750 HD			STANDARD		
	Optima CT660					
SIEMENS	Sensation 16			B31f		
	SOMATOM Definition AS					
	SOMATOM Definition Flash			I30f, I40f		
	SOMATOM Force			Br40d		

**Table 2 cancers-13-04638-t002:** Patient characteristics.

Characteristic	N = 61
Gender, male (%)	50 (81.9%)
Age, median (range)	66 (48–81)
Cirrhosis, yes (%)	56 (91.8%)
Child–Pugh grade	
A	56 (91.8%)
B	5 (8.2%)
Diameter of largest lesion, in mm, median (range)	37 (10–220)
Portal vein invasion, yes (%)	11 (18.1%)
Extrahepatic disease yes (%)	7 (11.4%)
* BCLC staging	
A	22 (36.1%)
B	22 (36.1%)
C	17 (27.8%)
** ECOG performance	
0	58 (95.1%)
1	3 (4.9%)

* Barcelona clinic liver cancer (BCLC) staging; ** European Cooperative Oncology Group (ECOG) performance.

## Data Availability

The data is privately owned by the trial coordinators.

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
