# Peer review of "Reproducibility of CT-Based Hepatocellular Carcinoma Radiomic Features across Different Contrast Imaging Phases: A Proof of Concept on SORAMIC Trial Data"

_cancers, 2021, doi:10.3390/cancers13184638_

Round 1
Reviewer 1 Report
The revised version of the manuscript has been improved.
Author Response
We thank the reviewer for the feedback that ultimately helped to improve the manuscript.
Reviewer 2 Report
Thank you for the opportunity to review the revised manuscript. I still believe that the manuscript is of interest to the scientific community and after revising, the manuscript should be ready for publication.
Author Response

(The authors gave the same response as above.)

Reviewer 3 Report
The authors performed a revision of their methodology and addressed comments from the previous review stage.
Thank you for open-sourcing the code. Please change the link to https://github.com/AbdallaIbrahim/The-reproducibility-and-ComBatability-of-Radiomic-features
I do not see some analyses her, please add them to the repository if possible. For instance, consider adding volume correlation analysis and the <0.85 correlation from the newely added sentence: "Of the concordant 22 shape features, 8 features were highly correlated with volume (R>0.85), in addition to 1 feature from the NGLDM group (NGLDM_DN) and 2 features from the GLRLM group (GLRLM_RLN and GLRLM_GLN). The remaining features (31,73.8%) had a correlation coefficient <0.85."
Author Response
We thank the reviewer for the feedback that ultimately helped to improve the manuscript.
Comment #1: "Thank you for open-sourcing the code. Please change the link to https://github.com/AbdallaIbrahim/The-reproducibility-and-ComBatability-of-Radiomic-features"
Our response: We added brackets to the link in the text:
"The analysis code used in this study can be found on: (https://github.com/AbdallaIbrahim/The-reproducibility-and-ComBatability-of-Radiomic-features)."
Comment #2: " I do not see some analyses her, please add them to the repository if possible. For instance, consider adding volume correlation analysis and the <0.85 correlation from the newely added sentence: "Of the concordant 22 shape features, 8 features were highly correlated with volume (R>0.85), in addition to 1 feature from the NGLDM group (NGLDM_DN) and 2 features from the GLRLM group (GLRLM_RLN and GLRLM_GLN). The remaining features (31,73.8%) had a correlation coefficient <0.85.""
Our response: We added the script "Assessing_correlation_with_volume" to the online repository.
Reviewer 4 Report
Thank you for your efforts to make a revision. The queries I pointed out were fully resolved. Thank you.
Author Response
We thank the reviewer for the feedback that ultimately helped to improve the manuscript.
This manuscript is a resubmission of an earlier submission. The following is a list of the peer review reports and author responses from that submission.
Round 1
Reviewer 1 Report
In this study the authors investigate in HCC patients the reproducibility of handcrafted radiomic features (HRFs) derived from arterial and portal venous phases CE-CT scans acquired with the same scanning parameters.
In general the reported data are poorly presented and described, and are not always clear. Tables and Figures are not properly reported, figure legends are not informative and do not provide useful information about the showed data.
Specific comments:
1) 104 ROIs were selected on patient scans and from these ROIs 167 HRFs were extracted. This difference (104 ROIs vs. 167 HRFs) should be commented. Whether multiple HRFs are obtained from the same ROI this should be clearly reported. Whether specific scanners produce multiple HRFs for the same ROI, this should be clearly described.
2)Poor description of the HRFs. Even thought a reference is included (reference 24) stating previous features description, this does not provide complete description of the HRFs reported in the present study. All the reported features should be properly described in the present study as well as the specific parameters of the HRFs; these latter are clearly showed in Figure 3 and 4 but not described in any part of the work.
3)The authors state “Looking at the concordance of HRFs (...), shape features had the highest concordance values..”. Why does this HRF, that appears to be the most relevant, is not present in Figure 3?
4) What CCC values are reported in Figure 3 and 4? Is it mean or median? This should be reported in Figure legend
5)Data before and after ComBat harmonization (Figure 3 and 4) are unnecessarily separated in the 2 figures. For a more clear comparison they should be showed in the same panel. The same observation for Table 3 and 4.
In addition, Table 3 and 4 could be more easily understandable and informative if represented by graphs, instead of simple numerical values. For instance by boxplots combined with dotplots to visualize also n samples distribution. Tabular data should be either included in addition to graphs or as supplementary files.
F)Table 2 poor quality:
-Poor table formatting. Row headings are not properly reported, parentheses should describe what is actually reported: “Gender (male) “ should be reported as “Gender, male (%)”; “Age (median,range)” should be reported as “Age, median (range)” and so on for all row headings.
Following the inappropriate row heading, a mistake is reported: the median age indicated in table is 66 and not 48 years as stated in the main text (48 appears to be the lower age range value)
-Other patient’ characteristics are showed but they are neither commented nor clearly understandable, acronyms should be defined in the maintext or in table notes.
G)Typos in section 2.3 hRF instead of HRF
Author Response
We would like to thank the editor and reviewers for their comments, which have ultimately led to an overall improvement of the manuscript. We have responded to each comment separately, and amended the manuscript accordingly. Please find our responses below. Line numbers refer to the revised manuscript.
Comment #1: “In this study the authors investigate in HCC patients the reproducibility of handcrafted radiomic features (HRFs) derived from arterial and portal venous phases CE-CT scans acquired with the same scanning parameters.
In general the reported data are poorly presented and described, and are not always clear. Tables and Figures are not properly reported, figure legends are not informative and do not provide useful information about the showed data.
Our response: We thank the reviewer for the comments and time spent reviewing the manuscript. While we disagree with the generalization that the data are poorly reported and described, we tried to address the points where misunderstandings could arise.
104 ROIs were selected on patient scans and from these ROIs 167 HRFs were extracted. This difference (104 ROIs vs. 167 HRFs) should be commented. Whether multiple HRFs are obtained from the same ROI this should be clearly reported. Whether specific scanners produce multiple HRFs for the same ROI, this should be clearly described.”
Our response: The number of handcrafted radiomic features is the same for all ROIs, regardless of the scanner used. We modified the sentence to be more clear:
“A total of 167 original HRFs were extracted from each of the available 104 ROIs.”
Comment #2: Poor description of the HRFs. Even thought a reference is included (reference 24) stating previous features description, this does not provide complete description of the HRFs reported in the present study. All the reported features should be properly described in the present study as well as the specific parameters of the HRFs; these latter are clearly showed in Figure 3 and 4 but not described in any part of the work.
Our response: We politely disagree. The extracted feature definitions are the same as previously published. We have described the settings used for feature extraction in lines:
“Image intensities were binned with a binwidth of 25 Hounsfield Units (HUs) in order to reduce noise levels and to reduce texture matrix sizes, and therewith computation power, with no resampling or further preprocessing of the images.” Figures 3 and 4 show the concordance correlation coefficient for the different features.
Comment #3: “The authors state “Looking at the concordance of HRFs (...), shape features had the highest concordance values..”. Why does this HRF, that appears to be the most relevant, is not present in Figure 3?”
Our response: Shape features are expected to have high concordance since we used the exact same segmentation on both scans. We did not include the shape features in the domain translation process. Therefore, to make both figures comparable we did not include them in the figures.
We mentioned in the discussion: “On the other hand, also according to expectations, HRFs that do not depend on the intensity values, but the shape of the segmentation (shape features), were found to be reproducible across both phases, with the exception of the shape feature centroid distance, which is based on the distribution of intensity values around the geometric center of the ROI.”
We also clarified in the results section: “Out of the 167 extracted HRFs, 42 (25%) were reproducible (had a CCC>0.9) across both phases (Figure 3a, shape features were not included to ease the comparison between figures).”
Comment #4: “What CCC values are reported in Figure 3 and 4? Is it mean or median? This should be reported in Figure legend”
Our response: The reported CCC value is the exact value as described in the legend, not the mean or median: “Figure 3. (a) The CCC values for the different HRFs before ComBat harmonization; (b) The CCC values for the different HRFs after ComBat harmonization”
Comment #5: “Data before and after ComBat harmonization (Figure 3 and 4) are unnecessarily separated in the 2 figures. For a more clear comparison they should be showed in the same panel. The same observation for Table 3 and 4.
In addition, Table 3 and 4 could be more easily understandable and informative if represented by graphs, instead of simple numerical values. For instance by boxplots combined with dotplots to visualize also n samples distribution. Tabular data should be either included in addition to graphs or as supplementary files.”
Our response: We merged the figures into figures 3a and 3b. Tables 3 and 4 have been removed.
Comment #6: “Poor table formatting. Row headings are not properly reported, parentheses should describe what is actually reported: “Gender (male) “ should be reported as “Gender, male (%)”; “Age (median,range)” should be reported as “Age, median (range)” and so on for all row headings.
Following the inappropriate row heading, a mistake is reported: the median age indicated in table is 66 and not 48 years as stated in the main text (48 appears to be the lower age range value)”
Our response: Table 2 has been adjusted according to the comment. Indeed, 48 is the lower range value. The text has been adjusted.
“The patients included (n=61) had a median age of 66 years.”
Comment #7: “Other patient’ characteristics are showed but they are neither commented nor clearly understandable, acronyms should be defined in the maintext or in table notes.”
Our response: The acronyms have been defined in the table notes.
Comment #8: “Typos in section 2.3 hRF instead of HRF”
Our response: The typo has been corrected.
Reviewer 2 Report
Thank you for the opportunity to review this interesting manuscript investigating the reproducibility of handcrafted radiomics features (HRF) between different contrast phases in HCC patients. The authors also explored the use of ComBat harmonization to increase comparability between phases. Their study showed that only HRFs related to tumor shape were also comparable between different contrast phases. The use of ComBat yielded only a slight advantage.
The study presented in this manuscript is of high value from a radiological perspective, as the implementation of radiomics in clinical practice has many hurdles to overcome. The rigorous selection of scans included increases comparability and should therefore be less prone to methodical errors. Unfortunately, due to this strict selection, the number of examinations is comparatively low. Nevertheless, the study is methodologically very well conducted. The use of ComBat harmonization to increase comparability is very interesting, even if its success has been marginal. The results highlight the influence of contrast phases on a radiomics-based evaluation. The manuscript can serve as a basis for decision-making for further studies and thus provides a valuable contribution to the possible implementation of radiomics in clinical practice and the design of further studies.
Author Response
We would like to thank the editor and reviewers for their comments, which have ultimately led to an overall improvement of the manuscript. We have responded to each comment separately, and amended the manuscript accordingly. Please find our responses below.
Comment #1: “Thank you for the opportunity to review this interesting manuscript investigating the reproducibility of handcrafted radiomics features (HRF) between different contrast phases in HCC patients. The authors also explored the use of ComBat harmonization to increase comparability between phases. Their study showed that only HRFs related to tumor shape were also comparable between different contrast phases. The use of ComBat yielded only a slight advantage.
The study presented in this manuscript is of high value from a radiological perspective, as the implementation of radiomics in clinical practice has many hurdles to overcome. The rigorous selection of scans included increases comparability and should therefore be less prone to methodical errors. Unfortunately, due to this strict selection, the number of examinations is comparatively low. Nevertheless, the study is methodologically very well conducted. The use of ComBat harmonization to increase comparability is very interesting, even if its success has been marginal. The results highlight the influence of contrast phases on a radiomics-based evaluation. The manuscript can serve as a basis for decision-making for further studies and thus provides a valuable contribution to the possible implementation of radiomics in clinical practice and the design of further studies.”
Our response: We thank the reviewer for the comment and support.
Reviewer 3 Report
Please see suggestions in the file attached

Author Response
We would like to thank the editor and reviewers for their comments, which have ultimately led to an overall improvement of the manuscript. We have responded to each comment separately, and amended the manuscript accordingly. Please find our responses below.
- General
“The authors propose a novel domain-translation application between two phases of contrast-enhanced computed tomography (CECT) for radiomics. The data of the SORAMIC liver cancer trial was used to run the experiments. The CECT images were manually delineated by two specialists and then used for radiomics extraction. The authors listed extraction settings and scanning protocols, which is considered to be an excellent practice for radiomic studies and, especially, radiomics harmonization studies.
I would be pleased if the authors would address some of my concerns listed below. In general, I consider this work of a high quality with a need of few minor improvements to be added. I would greatly encourage the authors to publish their statistical analyses and code as supplementary materials in their subsequent submission. This will help to further enhance radiomics analyses standardization.”
Comment #1: “Main concerns. I would first argue that different contrast-enhanced CT modes should probably considered as different modalities as they help to spot different physical effects. There is still a need in translating one modality to another (e.g. MR-only radiotherapy MR -> CT translation), so I would use the term “domain translation” instead of “harmonization” here. When the same CT scanner acquired two different non-contrast images with different scanning protocol this is a case of image harmonization. When contrast agent is used in arterial and venous phases, it produces two different image modalities, as seen in Figure 2.”
Our response: We thank the reviewer for the comments and time spent reviewing the manuscript. We agree, the term “harmonization” has been replaced with “domain translation”
Comment #2: “The most reproducible features are very much possibly to be ROI volume-confounded, as Welch et al pointed earlier. So I would add this substantial check to your analysis section (e.g. provide also Spearman correlation with ROI volume). This will probably leave ROI volume correlated features with CCC > 0.9. Please also add more discussion on this issue.”
Our response: Features’ correlation with volume has been assessed and reported according to the suggestion. The majority of the shape features were highly correlated with volume. However, only 3 features from the other feature groups had a Pearson correlation of 0.85. The remaining features were not found to be highly correlated with volume. We added the following to the manuscript:
Methods: “The correlation of concordant features with volume was assessed using Pearson correlation. Features that had a correlation coefficient > 0.85 were considered highly correlated.”
Results: “Of the concordant 22 shape features, 8 features were highly correlated with volume (R>0.85), in addition to 1 feature from the NGLDM group (NGLDM_DN) and 2 features from the GLRLM group (GLRLM_RLN and GLRLM_GLN). The remaining features (31, 73.8%) had a correlation coefficient <0.85.”
“Following ComBat harmonization, the number of highly correlated features with volume increased by one feature (Stats_energy). The concordant features before domain translation maintained their correlation with the volume.”
Discussion: “The correlation of radiomic features with the volume of the ROI has been considered one of the major points to be assessed in radiomics analysis, since some of the features were reported previously to be surrogates of volume [43]. In our analysis, we observed that the majority of the features identified as concordant (domain-translatable with ComBat) between the arterial and venous CT scans was considerable, most of which were shape features. However, the majority of features were not found to be highly correlated with volume, which means that these features can decode additional information about the ROIs being investigated.”
Comment #3: “Page 2, abstract: The authors state: “…, yet the number of HRFs could be used interchangeably between those phases” – this goes to the above-mentioned issue of volume-confounding effect in radiomics. Please check this effect and I would recommend an addition “…, yet the number of volume-confounded HRFs could be used interchangeably between those phases”.”
Our response: Based on the analysis, only 8 features from the shape group (n=22) and 3 features from the remaining features (n=3) were highly correlated with volume. Therefore, we did not generalize. We have added the volume correlation analysis to the manuscript as described above.
Comment #4: “Page 3, introduction: Please elaborate more on 1) what is CECT and its two phases, 2) why you want to domain-translate between these two different phases. Is that for augmenting the data when either of phases missing? On the other hand, you can get skepticism from some stating: these are two physically different modalities, they must show different things – why trying to map one to another. Try to address this criticism here also.”
Our response: We have added a brief description of the CECT and its phases in the introduction:
“Hepatocellular carcinoma (HCC) is the most common primary liver cancer, the fifth most common malignancy worldwide, and a leading cause of cancer-related mortality [13]. Different diagnostic approaches and treatment modalities are used clinically depending on the characteristics of the patient and the progression of the disease [14,15]. Contrast-enhanced computed tomography (CE-CT) scans are considered one of the main diagnostic tools for HCC. CE-CT can be acquired at different times following the injection of the contrast agent to acquire arterial, venous or late phase scans. Each phase shows specific characteristics for HCC lesions.”
We have also tried to further clarify the justification for this study:
“We hypothesize that the time of acquisition after the injection of the contrast agents adds another level of complexity to be accounted for in the radiomics analysis, as HRFs might be affected by the appearance of contrast, due to the variations in the distribution of the contrast within the lesions. As a proof of concept, we investigate the sensitivity of HRFs extracted from CE-CT scans depicting HCC acquired during the arterial and portal venous phases, when all other acquisition and reconstruction parameters were fixed. Furthermore, we investigate the potential of the ComBat method harmonization to for domain translation of the HRFs extracted from these scans. Ultimately, we aim to (i) guide the identification of HRFs that can be used interchangeably between arterial and venous phase scans, which could increase the number of scans that can be included in a CE-CT based radiomics study; and (ii) identify the features that can be used in studies analyzing both phases simultaneously to maximize the information extracted from ROIs.”
Comment #5: “Page 3, section 2.1: Please elaborate more on data selection. What do you mean by “Patients with poor quality CT imaging (n=48)”? What was considered “poor”?”
Our response: We have added the sentence to make it clearer:
“Scans that contained artifacts were considered of poor quality (n=48)”
Comment #6: “Page 4, table 1: Two issues regarding the table:
- Do I understand correctly: each scanner used only one unique kernel for reconstruction? What about the last two SOMATOM scanners? Did they have I40f also?
- Could you please elaborate on how you dealt with various slice thicknesses? Did you perform any additional resampling of some sort or extracted radiomics straight away after 25 HU binning?”
Our response: We adjusted the layout of table 1 to be clearer. The comparison was made between the arterial and venous phases for each patient. To avoid image resampling and other image preprocessing steps that could have an effect on the reproducibility of features, the scans included for each patient had the exact same settings including slice thickness. We extracted the features directly after the HU binning without resampling:
“From the available 338 patients with both arterial and portal venous scans available, patients with scans that had any difference in the acquisition or reconstruction parameters, or lacked segmentations reviewed by an expert, were excluded. A total of 61 patients with 104 distinct lesions were finally included in this study (Figure 1).”
Comment #7: “Page 6, section 3.3: The authors state: “Stats features were found to be the least concordant group…” – this might be also due to their relatively low correlation with ROI volume. Interestingly, the most concordant statistical feature Energy is a sum of each gray level squared and, oftentimes, has medium to very high correlation with the number of voxels and volume.”
Our response: We agree with the reviewer. Tables 3 and 4 have been removed.
Comment #8: “Page 7, section 3.4: Could authors provide some explanation why High Gray Level Zone Emphasis improves its concordance after ComBat? Is there any mathematical explanation. This is optional though.”
Our response: ComBat harmonization is an empirical bayes method. Therefore, we are not fully aware of why it would perform better on some features than others.
Comment # 9: “Pages 7-9, figures 3,4: As I addressed before, the concordance of many features, like GLRLM GLNU might be caused by its close relation to volume. Could you maybe highlight volume-confounded features differently? You can also highlight the feature name, for example.”
Our response: We have investigated and reported on the volume confounded features as previously mentioned.
Comment # 10: “Page 11, discussion: The authors state: “The application … did not significantly improve… of ComBat harmonization”. This can be dangerous to say. First of all, what was the metric of significance? The CCC > 0.9 was chosen manually, although it is a reasonable threshold, still many features’ CCC grew after the ComBat harmonization – was not this increase in CCC statistically significant? I would recommend revisiting this paragraph.”
Our response: We have rephrased the sentences to read:
“The number of features that had a CCC value higher than 0.9 was slightly higher after the application of ComBat on the HRFs extracted from the arterial and portal venous phases. ComBat successfully harmonized two additional HRFs compared to the number of concordant HRFs before domain translation.”
Comment # 11: “Page 1, Simple Summary: The text is nice and clear in general, but sometimes you see minor editing is still needed. For example “–the quantitative features extracted from medical images-” >> “–[space]the quantitative features extracted from medical images[space] –[long dash]”. I will not address grammar or style anymore – please check within the authors carefully once more prior to submission.”
Our response: Style and grammar have been checked and edited where needed.
Comment # 12: “Page 1, Simple Summary: The authors write “Furthermore, radiomics analyses require big data to achieve desirable performances.” I would argue that end-to-end convolutional neural nets need larger sample size (>1000 patients) to accommodate all the variance as the features are learned from scratch, whereas radiomics utilizes human prior knowledge, so it needs smaller sample size compare to the CNNs. If you have, for example, five highly predictive orthogonal features, you do not really need “big data”: 100-200 patient cases will allow to build nicely tuned and cross-validated inear algorithm or a tree. The problem is not data quantity, but quality, in my opinion.”
Our response: We agree that deep learning algorithms need larger datasets compared to handcrafted radiomics. However, the larger the number of included models, the more generalizable the radiomic models. Data quality remains a major issue for both methods. We added the following to the summary:
“Furthermore, radiomics analyses require big data with good quality to achieve desirable performances.”
Comment # 13: “Pages 4-5, figure 2: caption is moved to the next page. I guess that is more on a journal’s side.”
”Page 5, sections 2.3-2.4: I would encourage the authors to submit the code for a review in their subsequent submission. This will further enhance your analyses and will benefit the field in general.”
Our response: The code has been added in a github repository and referred to in the methods:
“The analysis code used in this study can be found on (https://github.com/AbdallaIbrahim/The-reproducibility-and-ComBatability-of-Radiomic-features).”
Comment # 14: “Tables 3, 4: Maybe these tables can be omitted as they somehow duplicate figures 3 and 4. I have some minor concerns regarding grouping features by their parent matrix or class. Although this was also done in the original Aerts et al study and seems to be intuitively logical, there are several issues to address. I would argue, if you take a GLRLM feature (say GLNU) that there are many features from other classes closer (a.k.a. more correlated to GLRLM GLNU) than some other GLRLM features. This way, grouping them by hierarchical clustering (with Spearman correlation as a distance metric, for instance) would group them better with regards to their relation to scanner protocol.”
Our response: Tables 3 and 4 have been omitted.
Comment # 15: “Figures 3, 4 – The feature names are cut on the left side. Consider giving more space for the feature names. I assume, the picture was snipped from Jupyter notebook, it looks grainy. If yes, please consider “open another view for the output” in Jupyter Lab, then you can export the image from there in higher quality than the snapshot.”
Our response: We have improved the quality of the figures.
Reviewer 4 Report
Thank you for giving us the opportunity to review your article. The authors investigated the application and efficacy of Combat harmonization to overcome the batch effect of CT images among multiple institutions. Although the results are not positive, the concept of this study is good, because collecting many images from multiple institutions is important to perform the study of radiomics.
I made a few comments. And I hope to improve your article.
Hepatocellular carcinoma (HCC) does not show the same enhancement features in the arterial and portal phase. For example, a tumor shows heterogenous enhancement in the arterial phase, and homogenous hypo enhancement in the portal phase. I cannot understand to why the authors compared these two phase. In my opinion, reproducibility between two images in the same phase, in the different cross section might be more suitable for the author’s objective.
Please describe the diagnostic criteria of HCC. Based on images? Pathology?
Author Response
We would like to thank the editor and reviewers for their comments, which have ultimately led to an overall improvement of the manuscript. We have responded to each comment separately, and amended the manuscript accordingly. Please find our responses below.
Comment #1: “Thank you for giving us the opportunity to review your article. The authors investigated the application and efficacy of Combat harmonization to overcome the batch effect of CT images among multiple institutions. Although the results are not positive, the concept of this study is good, because collecting many images from multiple institutions is important to perform the study of radiomics.
I made a few comments. And I hope to improve your article.
Hepatocellular carcinoma (HCC) does not show the same enhancement features in the arterial and portal phase. For example, a tumor shows heterogenous enhancement in the arterial phase, and homogenous hypo enhancement in the portal phase. I cannot understand to why the authors compared these two phase. In my opinion, reproducibility between two images in the same phase, in the different cross section might be more suitable for the author’s objective.”
Our response: We thank the reviewer for the comments and time spent reviewing the manuscript. While we agree that the reproducibility of features within the same phase is of great importance, such data are unfortunately not available to us. We tried to make the justification for this study clearer in the introduction:
“We hypothesize that the time of acquisition after the injection of the contrast agents adds another level of complexity to be accounted for in the radiomics analysis, as HRFs might be affected by the appearance of contrast, due to the variations in the distribution of the contrast within the lesions. As a proof of concept, we investigate the sensitivity of HRFs extracted from CE-CT scans depicting HCC acquired during the arterial and portal venous phases, when all other acquisition and reconstruction parameters were fixed. Furthermore, we investigate the potential of the ComBat harmonization for domain translation of the HRFs extracted from these scans. Ultimately, we aim to (i) guide the identification of HRFs that can be used interchangeably between arterial and venous phase scans, which could increase the number of scans that can be included in a CE-CT based radiomics study; and (ii) identify the features that can be used in studies analyzing both phases simultaneously to maximize the information extracted from ROIs.”
Comment #2: “Please describe the diagnostic criteria of HCC. Based on images? Pathology?”
Our response: We have added the diagnostic criteria used in the methods section:
“Imaging data for 424 patients diagnosed with HCC (using cyto-histological criteria, radiologic criteria, or a combination of both) were obtained for the SORAMIC trial, of which 338 scans were available for analysis in this study.”